# Start Codon Recognition in Eukaryotic and Archaeal Translation Initiation: A Common Structural Core

**DOI:** 10.3390/ijms20040939

**Published:** 2019-02-21

**Authors:** Emmanuelle Schmitt, Pierre-Damien Coureux, Auriane Monestier, Etienne Dubiez, Yves Mechulam

**Affiliations:** Laboratoire de Biochimie, Ecole polytechnique, CNRS, Université Paris-Saclay, 91128 Palaiseau CEDEX, France; pierre-damien.coureux@polytechnique.edu (P.-D.C.); auriane.monestier@inra.fr (A.M.); Etienne.dubiez@ed.ac.uk (E.D.)

**Keywords:** archaea, eukaryotes, ribosome, translation initiation, evolution

## Abstract

Understanding molecular mechanisms of ribosomal translation sheds light on the emergence and evolution of protein synthesis in the three domains of life. Universally, ribosomal translation is described in three steps: initiation, elongation and termination. During initiation, a macromolecular complex assembled around the small ribosomal subunit selects the start codon on the mRNA and defines the open reading frame. In this review, we focus on the comparison of start codon selection mechanisms in eukaryotes and archaea. Eukaryotic translation initiation is a very complicated process, involving many initiation factors. The most widespread mechanism for the discovery of the start codon is the scanning of the mRNA by a pre-initiation complex until the first AUG codon in a correct context is found. In archaea, long-range scanning does not occur because of the presence of Shine-Dalgarno (SD) sequences or of short 5′ untranslated regions. However, archaeal and eukaryotic translation initiations have three initiation factors in common: e/aIF1, e/aIF1A and e/aIF2 are directly involved in the selection of the start codon. Therefore, the idea that these archaeal and eukaryotic factors fulfill similar functions within a common structural ribosomal core complex has emerged. A divergence between eukaryotic and archaeal factors allowed for the adaptation to the long-range scanning process versus the SD mediated prepositioning of the ribosome.

## 1. Introduction

In the 1980s, pioneering studies demonstrated that sequence analysis and molecular structures revealed evolutionary relationships between organisms. From this period, the sequences of 16S/18S rRNA genes, present in all organisms and slowly evolving, have been used as phylogenetic markers [1]. These works led to the definition of a new taxon called a “domain” and to the division of the tree of life into three monophyletic domains, bacteria, archaea and eukarya, each comprising several kingdoms [2,3]. The root of this tree separates bacteria from the two other domains, archaea and eukarya, that appear as distant relatives. Indeed, most informational genes in archaea, in particular those related to ribosomal translation (but also those involved in DNA replication and transcription), are closer to their eukaryotic counterparts than to their bacterial ones [4,5,6]). During the same period, another theory has emerged, first from electron microscopy studies of ribosomes from eocytes (nowadays called crenarchaeotes) and also from the use of alternative strategies for phylogenetic reconstructions [7,8]. This led to a two-domain tree, called the eocyte tree, with eukaryotes originating from within an archaeal phylum instead of being a sister lineage, in such a way that archaea would be paraphyletic [9,10]. The use of new statistical methods for studying phylogeny [11,12,13] and recent developments of metagenomics or of single-cell genomics [14,15] further reinforce this theory [16,17,18,19]. This model is still a matter of debate and other trees of life are defended (see for instance [14,16,17,18,19,20,21]). In particular, the monophyly of archaea is privileged in a two-domain tree updated from [3] where archaea and eukaryotes are members of a clade named “Arkarya” [20]. In an alternative two-domain model, archaea and bacteria are sister clades, sharing a common “akaryote” ancestor to the exclusion of eukaryotes, as previously proposed [22,23,24]. This latter model emerged from reconstructions based on the genome content of coding sequences for protein domains [25,26]. Overall, depending on the model used for the tree of life, reductive evolution processes may have played a role [23,26,27]. As pointed out in [20], it is important to focus on biological plausibility and on comparative molecular biology in order to help resolve phylogenetic impasses. For instance, the sisterhood of eukaryotes and archaea appears supported by the proximity of the archaeal and eukaryotic ribosomes and further confirmed by structural studies showing that archaeal ribosomal proteins are either universal or shared with eukaryotes [28]. These new developments strengthen the importance of studying molecular mechanisms of fundamental processes in archaeal organisms, such as ribosomal translation, not only because they are interesting as such but also because their studies can help to better understand phylogenetic links between archaea and eukaryotes. Finally, because archaeal functional systems are usually simpler than their counterparts in eukaryotes, biochemical and structural studies in archaea can also provide valuable models of eukaryotic processes [29,30,31,32,33,34].

This review focuses on molecular mechanisms of protein synthesis initiation. Universally, this step allows for an accurate selection of the start codon on mRNA and consequently the definition of the open reading frame. In eukaryotes, this process is very complicated with many initiation factors involved, and it is therefore the target of many regulations (see for instance [35,36]). The mechanism involves a pre-initiation complex (PIC) made up of the small ribosomal subunit bound to the ternary complex eIF2-GTP-Met-tRNA_i_^Met^ (TC), the two small initiation factors eIF1 and eIF1A, as well as two proteins that have a more regulatory function in the process. These are eIF5, the guanine activating protein (GAP) of eIF2 and the multimeric factor eIF3. In the presence of factors belonging to the eIF4 family, the pre-initiation ribosomal complex (PIC) is recruited near the 5′-capped end and scans the mRNA until an AUG codon (frequently the first) in a correct context (Kozak motif) is found. The AUG recognition stops the scanning, provokes a factor release and the assembly of an elongation-proficient 80S complex through large subunit joining, with the help of eIF5B and eIF1A (Figure 1). Beside this canonical mechanism, a number of alternative initiation routes have been described (Table 1). This includes non-canonical cap-dependent translation initiations as exemplified in [37], entry of the ribosome mediated by specific mRNA structures such as IRES (internal ribosome entry site) [38], cap-independent translation enhancers [39] or eIF4F-resistant translation initiation of N6-methyladenosine-containing mRNA under stress conditions [40]. In addition, mRNAs lacking 5′ untranslated regions (leaderless mRNAs) are translated by various non-canonical mechanisms [41].

In archaea, there is no long-range scanning because mRNAs have Shine-Dalgarno sequences or very short 5′ UTR (Table 1). However, genomic analyses have shown that three initiation factors homologous to their eukaryotic counterparts, aIF1, aIF1A and aIF2, are found ([5,6,42], Figure 1 and Table 1). Thus, even if obvious differences between eukaryotes and archaea exist, in particular for the recruitment of the PIC, the start codon selection is achieved within a common structural core made up of the small ribosomal subunit, the mRNA, the methionine initiator tRNA (Met-tRNA_i_^Met^) and the three initiation factors e/aIF1, e/aIF1A and e/aIF2 (Figure 2).

This review gathers data from the structural and functional studies of the three eukaryotic and archaeal initiation factors, e/aIF1, e/aIF1A and e/aIF2. The scanning mechanism in eukaryotes and the SD-dependent mechanism in archaea are mainly considered (Figure 1). The similarities and divergences of the initiation factors in both domains are highlighted. Although eukaryotic factors have been studied for a long time, the studies of their archaeal representatives had been sparser. Still, they have brought valuable information on understanding eukaryotic translation initiation mechanisms. Moreover, recent structural studies allowed a description of molecular complexes during the eukaryotic and archaeal translation initiation. At a time when the idea that archaea can be the ancestors of eukaryotes is reinforced, these data raise new elements to discuss the molecular evolution of the translation initiation processes.

## 2. Features of Eukaryotic and Archaeal Translation Initiation Factors

### 2.1. e/aIF2

In both domains of life, the Met-initiator tRNA is carried to the small ribosomal subunit by the heterotrimeric (α, β and γ) GTP-binding factor e/aIF2 [45]. The factor is specific to the initiator tRNA, strongly recognizes the esterified methionine group and prefers an A_1_-U_72_ base pair, as found in all eukaryotic or archaeal initiator tRNAs [46,47,48,49,50,51]. Early on, the crucial role of eIF2 in the start codon selection was brought to light from the identification of yeast mutants able to initiate translation on a non-AUG codon [52,53,54,55,56,57]. Further studies showed that correct pairing between the start codon and the Met-tRNA_i_^Met^ anticodon induced the release from the ribosome of e/aIF2 in a GDP bound state. Therefore, the selection of the start codon is achieved through the control of the nucleotide state of e/aIF2 [58].

#### 2.1.1. e/aIF2-tRNA Complex

At the beginning of eIF2 studies, sequence alignments had suggested that the γ subunit shared homologies with the elongation factor EF1A, responsible for the handling of the aminoacylated elongator tRNA during the translation elongation [59,60]. Structural studies of the eukaryotic versions of the heterotrimer or of the isolated eIF2γ subunit had been hampered because of difficulties in purification. However, structural studies were successfully performed with archaeal versions of the protein. Numerous crystal structures of isolated subunits and finally of the full heterotrimeric protein were obtained [45,48,61,62,63,64,65,66]. Consistent with sequence alignment predictions, aIF2γ is a three-domain protein binding GTP-Mg^2+^ in its domain I, similar to EF1A. Conformations of two switch regions (switch 1 and 2) and of the GKT loop control the nucleotidic state of the factor [67]. aIF2α is made up of three domains (Figure 3, [68]). Its C-terminal domain is bound to the domain II of aIF2γ. aIF2β has a long N-terminal helix linked to a structural module containing an α–β domain and a zinc binding domain. The N-terminal helix alone ensures the anchoring of the β subunit to the nucleotidic domain of γ (Figure 3). In the heterotrimer, the peripheral α and β subunits do not interact. Notably, comparison of all available structures showed that within the trimer, there is a “rigid unit” (containing γ, the C-terminal domain of α and the N terminal helix of β) surrounded by two mobile wings: domains 1 and 2 of α, and the core domain of β [63,64,66]. In 2012, a 5 Å crystal structure of the TC (aIF2:GDPNP:Met-tRNA) was solved [69]. This structure showed that the initiator tRNA was essentially bound to aIF2 via the domain 3 of α and the domains I and II of γ, while the aIF2β subunit did not strongly contribute to the tRNA binding. Unexpectedly, the overall tRNA binding mode to aIF2γ was totally different from that observed for the binding of an elongator tRNA to EF1A [69,70]. The initiator tRNA binding mode by aIF2 observed in the crystal structure was further supported by SAXS experiments and by a Kd value determination using aIF2 variants [48,49,69,71].

At first glance, the use of similar tRNA binding modes by eukaryotic and archaeal e/aIF2 was not obvious. Indeed, early biochemical studies using yeast eIF2 had shown that the α subunit contributed only slightly to tRNA binding affinity [74]. Interestingly, eukaryotic α subunits specifically harbor an acidic C-terminal extension (Figure 3). The removal of this extension was sufficient to reveal the positive effect of the α subunit on the tRNA binding affinity in eukaryotic complexes [75]. This argued in favor of the same tRNA binding mode in both domains of life. It has been proposed that during the translation initiation in eukaryotes, the negative effect of the acidic tail might be relieved upon the interaction of the TC with the ribosome. Finally, an adjustment of the influence of each peripheral subunit on the tRNA binding affinity was observed depending on the e/aIF2 species studied [76]. This suggested a possible remodeling of the position of the mobile wings of e/aIF2 in the TC during the translation initiation, as discussed below. It can also be noted that another crystal structure of the TC was determined but with a very different conformation [77]. However, no confirmation of the biological significance of this conformation has been obtained yet.

#### 2.1.2. Nucleotide Cycle on e/aIF2

Other eukaryotic specificities such as added domains or extensions are deduced from sequence alignments (Figure 3). The most striking ones are those related to the nucleotide cycle on eIF2. In eukaryotes, two proteins assist eIF2: a guanine activating protein (GAP), eIF5 [58], and a guanine exchange factor (GEF), eIF2B [78,79]. eIF5 is monomeric and eIF2B is a heterodecameric complex (2 sets of α, β, γ, δ and ε subunits) with γ and ε having a catalytic function [80,81]. In the β subunit of eIF2, a eukaryote-specific N-terminal domain containing lysine-rich boxes was shown to be responsible for the binding of eIF5 and of the catalytic subunit of eIF2B (eIF2Bε) [82,83]. Moreover, the translation initiation is regulated via the phosphorylation of a conserved serine residue (S51 in yeast) on eIF2α (see, for instance, [84,85]). This phosphorylation leads to the formation of a highly stable complex between eIF2B and phosphorylated eIF2. Because cellular concentration of eIF2B is low compared to that of eIF2, this inhibits the exchange of GDP for GTP in eIF2 thereby preventing formation of TC and down-regulating the global translation level [86,87]. Interestingly, although the eIF2α phosphorylation decreases the level of translation of many genes, other genes that possess a short upstream open reading frame (5′uORFs) [88,89] or are IRES-dependent [90,91,92] are up-regulated. When the availability of TC is low, the stabilization of the ribosomal initiation complex at the start codon of the short uORFs is disfavored and the ribosome scanning is continued up to the ORF corresponding to the targeted protein (leaky scanning). In yeast and mammals, GCN4 and ATF4 are two examples of activated transcription factors acting as “master regulators” in response to amino acid starvation [93,94,95]. The whole regulation process triggered by the phosphorylation of eIF2α is termed the integrated stress response (ISR) [84,85]. Many studies have shown that translational regulations involve alternative routes for translation initiation using a particular set of initiation factors that can substitute for eIF2, such as MCT-1/DENR [96], eIF2D [97], eIF2A [98]. In addition, under some stress conditions such as hypoxia [99], eIF5B was proposed to participate in the recruitment of the initiator tRNA, in addition to its role in subunit joining [90,91,100]. This is reminiscent of the bacterial homologue of e/aIF5B, IF2, which in bacteria is involved in both initiator tRNA recruitment and subunit joining [101]. 

In archaea, no homologues of eIF5 and of the two catalytic subunits of eIF2B (γ and ε) are found. Therefore, the nucleotide cycle is thought to be non-assisted by a GAP and a GEF. Consistent with the absence of GEF, aIF2 from *Sulfolobus solfataricus* (Ss-aIF2) was found to have similar affinities for GDP and GTP [67,71]. Moreover, high-resolution crystal structures of Ss-aIF2γ in the presence of GTP-Mg^2+^ and molecular dynamics simulations suggested that in archaea a second magnesium ion could play a similar role as the one played in trans by the catalytic arginine residue of the GAP in eukaryotes [67]. In favor of this hypothesis, it is notable that the second magnesium ion is bound by two residues (in Ss-aIF2γ; D19 from the GKT loop and G44 from switch1) strictly conserved in all archaeal aIF2γ sequences whereas they are systematically replaced by A and N in eukaryotes [67]. 

Finally, in archaea, the equivalent of S51 is not strictly conserved in aIF2α. Phosphorylation at this position was reported with aIF2α from *Pyrococcus horikoshii* [102]. However no further confirmation of a possible role of this phosphorylation in aIF2 regulation was obtained, rather suggesting that translation is not regulated via aIF2α phosphorylation in archaea [103,104]. By analogy to the eukaryotic case where eIF5B can substitute for eIF2 (see above), it cannot be excluded that aIF2-independent processes in which aIF5B ensures tRNA recruitment as well as subunit joining, may also occur in specific conditions or for specific mRNAs.

#### 2.1.3. Regulation of eIF2 Assembly by Cdc123

In archaea, the assembly of the three subunits in the heterotrimeric form of the factor does not require any assistance. Accordingly, the three subunits can be purified independently and the heterotrimer assembled in vitro [48,49,71]. In *S. cerevisiae*, however, the essential eukaryote-specific Cdc123 protein was shown to be necessary to the formation of the eIF2 heterotrimeric complex, as it promotes the eIF2αγ assembly step [105,106,107]. Interestingly, the activity of Cdc123 requires a short eukaryote-specific C-terminal extension in eIF2γ ([106] and Figure 3). To date, Cdc123 orthologues appear widely distributed across all major lineages of the eukaryotic tree. Therefore, the phyletic pattern, together with its essentiality in yeast, indicates that Cdc123 might be a general regulator of translation initiation in eukaryotes [108]. 

### 2.2. e/aIF1A

The role of eIF1A (formerly called eIF4C) in the translation initiation was first evidenced in ribosomal complex preparations [109,110]. Eukaryotic sequences of eIF1A representatives showed a high-level of conservation and a “dipole nature” of the protein harboring a basic N-terminal extremity and an acidic C-terminal one [111,112]. The structure of human eIF1A was determined by NMR [113]. The protein contains an OB-fold with two C-terminal α-helices packed onto it. The N and C-terminal extensions are disordered. Archaeal aIF1A shares about 40% of sequence identity with its eukaryotic homologue. The 3D structures of e/aIF1A proteins are superimposable with very good rms values. For example, the structures of aIF1 from *P. abyssi* (PDB ID 4MNO) and eIF1A from *Cryptosporidium parvum* (PDB ID 2OQK) have an rms value of 0.64 Å for 72 compared C-alpha atoms. Notably, whereas the core of the protein is well conserved, the archaeal version of the factor does not possess an acidic C-terminal extension and has a shortened N-terminal basic tail (Figure 3). Hydroxyl radical probing, and then X-ray and cryo-EM structures, showed that eIF1A and aIF1A occupy similar positions on the small ribosomal subunit, in front of the A site, thereby precluding any elongator tRNA binding [114,115,116,117,118].

### 2.3. e/aIF1

Like eIF2, the role of eIF1 in the start codon selection was first defined genetically [119]. Since then, a huge amount of data has proven the crucial role of this factor in the dynamics and the accuracy of the eukaryotic translation initiation process (reviewed in [35]). e/aIF1 is a small protein of about 100 residues in archaea and 110 residues in eukaryotes (Figure 3). The 3D structure of human eIF1 was first determined using NMR [120]. The factor is made up of an α–β domain (29–113) and of an N-terminal unstructured domain (1–28). The archaeal proteins are highly similar to the eukaryotic proteins ([72,121] and Figure 3). The presence of a zinc knuckle in the N-domain is suggested for some archaeal representatives [72], but no 3D structure of this domain has been determined yet. Directed hydroxyl radical probing, and then crystallographic and cryo-EM structures, have shown the position of the factor on the small ribosomal subunit, in front of the P site on h44 [115,117,118,122,123].

## 3. The Three Initiation Factors in the PIC

### 3.1. The Scanning Model in Eukaryotes

The two small initiation factors eIF1 and eIF1A promote the formation of an open 48S complex competent for scanning [124,125]. eIF1A acts synergistically with eIF1 to bind the 40S subunit [126], and both factors facilitate the recruitment of the TC [127,128]. The acidic C-terminal tail of eIF1A contains two motifs, dubbed scanning enhancers (SE_1_ and SE_2_), important for mRNA scanning [129]. In contrast, the basic N-terminal extension of eIF1A has an opposite effect, with two regions (the scanning inhibitors SI1 and SI2) that disfavor the scanning-competent conformation of the ribosome. Together, these two regions of eIF1A favor the accurate accommodation of the initiator tRNA in the P site when an AUG codon is found [130,131,132]. Important advances on the mechanism of start codon selection have been obtained from kinetic studies using partially reconstituted yeast translation initiation systems [133]. Notably, the data have been obtained in the absence of eIF3, in a minimal eukaryotic initiation system close to the archaeal one. These studies showed that rather than via GTP hydrolysis by eIF2, the release of inorganic phosphate (Pi) from eIF2 controls the recognition of the start codon. This has led to a new description of the dynamics of the system. AUG recognition stops the scanning of the PIC, triggers the eIF1 departure, provokes the Pi release from eIF2, and then the further departure of eIF2-GDP. Consistent with this model, the full accommodation of the initiator tRNA in the P site upon the start codon recognition would cause a clash with eIF1 and therefore trigger its release, as suggested by cryo-EM studies [117,118,123]. 

Recent cryo-EM structures of partial yeast PIC (py-48S; 40S:mRNA:TC:eIF1:eIF1A:eIF5 and eIF3) further illustrate molecular events during the transition between the open scanning-competent conformation of the 48S complex and a closed conformation stalled on the start codon ([116,134], Figure 4). In the open scanning-competent state of the PIC, the mRNA is loosely bound in its channel and the initiator tRNA is not fully engaged in the P site. The P_OUT_ conformation of the tRNA appears to be stabilized around the anticodon loop by interactions with the core domain of eIF2β, and with eIF1 and eIF1A. Because the complex is scanning the mRNA, GTP hydrolysis on eIF2, activated by eIF5, can occur, but the Pi remains held onto eIF2. Base pairing in the P site induces a rotation of the head of the small ribosomal subunit towards the body, thereby closing the so-called latch region. This movement contributes to further stabilizing the mRNA in its channel, as well as the tRNA. In addition, the basic N-terminal tail of eIF1A favors the codon:anticodon interaction. Finally, the adjustment of the tRNA in the P site interferes with the eIF1 binding site at the level of its basic loop, in front of the anticodon, and at the level of loop 2, near the D-stem loop region of the tRNA. These close contacts disfavor the binding of eIF1 and trigger both its departure and the following events [73,135]. 

In the cryo-EM structures of eukaryotic PIC [116,134,136,137], the TC was modeled from the X-ray structure of its archaeal representative [69]. The rigid unit of eIF2 (γ:αD3:β:anchoring-helix) is bound to the acceptor helix of the tRNA as in the crystallographic structure of the archaeal TC, although the two mobile wings (the αD12 domain and the core domain of β) are repositioned. No contact exists between the eIF2 rigid unit, containing the GTP binding site, and the ribosome. However, in the P_IN_ closed state [116], eIF2αD1 is bound in the E site and interacts with the mRNA nucleotides -2 to -3, as previously proposed from hydroxyl radical probing experiments [138]. These interactions, which help stabilize the mRNA in its channel, are absent in the P_OUT_ conformation where mRNA is not visible in the E site (Figure 4). Moreover, the core module of β is seen in contact with the anticodon stem-loop of the P_OUT_ tRNA_i_, whereas it would rather interact with the upper part of the tRNA in the P_IN_ complex. Finally, the eukaryote-specific lysine-rich N-terminal domain of eIF2β is visible in neither of the two structures. Accordingly, the position of eIF5 could only be tentatively assigned. Altogether, these observations reinforce the idea that the versatility of the two mobile wings of e/aIF2 plays an important role in the dynamics of the translation initiation process. Intriguingly, no contact between eIF1 and eIF2 is seen in the available structures, and therefore the structural basis showing how the eIF1 departure triggers the release from eIF2 is still missing. The possibility that the “hub” protein eIF3 relays information from eIF1 to eIF2 is not excluded. Indeed, this factor participates in all of the initiation steps [139,140,141], and a close proximity between eIF3b, h44 and eIF2γ has been proposed from the cryo-EM structures ([134,142], Figure 4). Finally, recent work also suggested that the dynamics of the eIF5 interaction network also plays an important role in the release of eIF2-GDP [143]. 

### 3.2. Control of AUG Selection by aIF2 in Archaea

The full *Pyrococcus abyssi* translation pre-initiation complex (30S:mRNA:TC:GDPNP:aIF1:aIF1A) was reconstituted and studied by cryo-EM. The PIC contains the 30S, an mRNA with a strong SD sequence, the TC made with GDPNP, and the two small initiation factors aIF1 and aIF1A [115]. Two conformations of the full PIC were observed (Figure 4). The two small initiation factors aIF1 and aIF1A are bound at positions similar to their eukaryotic homologues. In the major conformation, dubbed PIC0-P_REMOTE_, the anticodon stem-loop of the tRNA is out of the P site (Figure 4). The TC structure is similar to that of the free TC, showing that the ribosome does not constrain the TC. In contrast to the eukaryotic PIC, the γ subunit is tightly bound to the 30S. The contacts involve the aIF2γ domain III (aIF2γ-DIII) and a long L2 loop of its domain II ([115] and Figure 4). Moreover, aIF2γ-DIII is bound to h44 and interacts with aIF1. In particular, the N-terminal domain of aIF1 contacts aIF2γ at the level of the two switch regions which control the nucleotide binding. This network of interactions was never observed before in eukaryotes or in archaea. Still, the contacts between eIF2γ-DIII and h44 had been suggested in the eukaryotic PIC using directed hydroxyl radical probing [144]. In the second conformation, called PIC1-P_IN_, the anticodon stem-loop of the initiator tRNA is bound within the P site, while the position of aIF2γ on h44 has not changed. Therefore, the structure of the TC is constrained but it is thought that the codon-anticodon pairing in the P site compensates for the structural constraint. Altogether, the data led to a novel view of the role of aIF2 in the start codon selection. The resulting model shows that during the archaeal translation initiation, the tRNA bound to aIF2 oscillates between the two positions observed in PIC0-P_REMOTE_ and PIC1-P_IN_ (Figure 5). These two positions are in equilibrium and the transition from one position to the other, modeled by superimposing PIC0 to PIC1, reflects the dynamics of the PIC during the testing for the presence of a start codon in the P site. The TC complex acts as a spring, pulling the initiator tRNA out of the P site as long as its position is not stabilized by a codon-anticodon pairing (Figure 5 and [115]). When a proper start codon is found, the base pairing with the initiator tRNA would stabilize the PIC1-P_IN_ conformation by compensating for the structural constraint on the TC. This would ensure a longer stay of the initiator tRNA in the P site and thereby trigger the aIF1 departure because of steric hindrance. The aIF1 release will also relieve contacts between aIF1 and the switch regions of aIF2γ. Therefore, it is tempting to imagine that the aIF1 departure causes the release of the Pi group from aIF2 and renders the process irreversible, as observed in eukaryotes [133]. The role of aIF1-induced dynamics of the PIC in the start codon selection was recently supported by toeprinting experiments [72]. The oscillation between the two conformations is favored by the presence of aIF1 which prevents a full tRNA accommodation through a competition for a same binding site (Figure 5). In the absence of aIF1, the PIC becomes more stable, as observed by a restricted toeprinting signal [72], and the tRNA is fully accommodated within the P site, as observed in a cryo-EM structure of a PIC devoid of aIF1 (to be published). Importantly, in this structure the anticodon stem-loop of the initiator tRNA is stabilized by interactions involving the C-terminal tail of three universal ribosomal proteins, illustrating a key role of the ribosome itself in the start codon selection.

Finally, it should be underlined that the two mobile wings of aIF2 also move during the search for a start codon in the P site (Figure 4). Their movements may contribute to the start codon selection process. Notably, the αD12 domain is never observed in the E site, in contrast to the eukaryotic case (Figure 4). Because in eukaryotes, the interaction involving αD12 and the mRNA plays a role in the selection of the start codon context [138], it is possible that the SD:antiSD interaction renders the interaction of αD12 in the E site unnecessary in this archaeal case. It is also interesting to note that aIF1A possesses an N-tail (Figure 3). Although shorter than that of its eukaryotic orthologue, the tail might also stabilize the codon-anticodon interaction in the P site. However, such an interaction has not yet been observed in the archaeal cryo-EM structures, and the question remains open.

## 4. Evolution of Translation Initiation Mechanisms and Concluding Remarks

The recent cryo-EM studies of archaeal and eukaryotic translation initiation complexes have shed light on their structural features in the two domains of life. Within a similar minimal translation initiation complex, the core domains of the three initiation factors 1, 1A and 2 fulfill similar functions. The archaeal system can be viewed as the simplest translation initiation process that involves the tRNA binding protein aIF2. The equilibrium between the PIC0-P_REMOTE_ and PIC1-P_IN_ conformations reflects a local scanning of the PIC to search for a start codon in the P site. aIF2 would play an active role in this process by pulling the tRNA out of the P site if no start codon was present. Such a mechanism can therefore appear as an ancestor of long-range scanning, as observed in eukaryotes. Strikingly, the emergence of the long-range scanning process along evolution is accompanied by the appearance of sequence specificities in the initiation factors, such as the well-characterized N and C extensions of eIF1A [131,132] or the L2 loop of eIF1 ([73], Figure 3). In e/aIF2, it appears that the two mobile wings have evolved differently in the two systems. Therefore, the position of eIF2αD12 in the E site seems to be related to the recognition of specific mRNA elements during scanning. Also, the role of the eukaryote-specific acidic C-terminal tail of eIF2α in the PIC remains to be determined. Starting from the minimal core complex, eukaryotes have also evolved towards more sophisticated processes that allow the emergence of additional regulation mechanisms. This is illustrated by the regulation of GTP hydrolysis by eIF5 and of GDP/GTP exchange on eIF2 by the GEF eIF2B. The available data are not sufficient to establish whether the active role of eIF2 in pulling the tRNA out of the P site also operates in eukaryotes. If this turned out to be the case, it would be another indication in favor of the emergence of eukaryotes from within an archaeal phylum.

Finally, the late step of translation initiation occurring after the e/aIF2-GDP departure involves the two e/aIF1A and e/aIF5B factors for a final check of the presence of an initiator tRNA and large subunit joining ([146,147,148], Figure 1). It is striking that two closely related factors, IF1 (homologous to e/aIF1A) and IF2 (homologous to e/aIF5B) also operate for large ribosomal subunit joining in bacteria [101,149,150]. This joining step therefore has a universal character (Table 1). Moreover, several studies have shown that in some cases eukaryotic translation initiation uses eIF5B instead of eIF2 to ensure the initiator tRNA recruitment [90,91,99,100]. This further argues in favor of an ancestral translation initiation mechanism involving e/aIF5B/IF2, with e/aIF1A/IF1 having relics in all domains of life and a possible representative of that being used in the LUCA. Evolution might then have selected the formylation of the initiator tRNA in bacteria to favor specificity, whereas this improvement would have been gained in eukaryotes and archaea thanks to the emergence of e/aIF2.

The evolution of translation initiation mechanisms is also related to the evolution of the 5′ untranslated regions of mRNA. For instance, mRNAs having very short or no 5′UTRs, called leaderless mRNAs, are present in all domains of life [151,152]. Such leaderless mRNAs are abundant in some archaea such as *S. solfataricus* [153,154,155,156] and *Haloferax volcanii* [157], though less abundant in others such as *P. abyssi* [158] and *Aeropyrum pernix* [153]. The translation initiation mechanism for the archaeal leaderless mRNA remains a matter of debate. In *S. solfataricus*, the formation of a stable complex between a leaderless mRNA and the 30S subunit was shown to depend on the presence of the initiator tRNA [104,159]. However, the requirement for initiation factors and even a possible start with 70S ribosomes remain to be studied in detail. The translation initiation of leaderless mRNAs is more documented in bacteria [151,160,161]. Although a favorable action of IF2 and an unfavorable one of IF3 were observed, both pathways involving either 30S or 70S IC are plausible, and further studies are required to determine which of the two routes is prevalent in vivo.

Finally, diversity in possible translation initiation pathways is also observed for leaderless mRNA translation in eukaryotes [41]. This raises the intriguing question of whether these leaderless mRNAs, possibly translated from an initiation mechanism involving the assembled 70S/80S ribosome, are relics of what happened in LUCA, or whether, on the contrary, they represent an evolved form of translation initiation allowing a diversification of the mechanisms for regulation. The same question is raised by the increasing number of alternative translation initiation routes that are being identified [37,41,96,162,163]. Overall, studies at the molecular level of the mechanisms of central biological processes have the potential to give important clues regarding the emergence and evolution of life as described in the present review (see, for instance, [164], [34] for another case study). A more recent example was given by the possible occurrence of a dynamic network of actin in Asgard archaea, which argues in favor of the proximity of these archaea to eukaryotes [165,166]. Information gained from biochemical studies can be used to complement phylogenetic data and to, at least, assess the likelihood of the possible evolution scenarios.

## Figures and Tables

**Figure 1 ijms-20-00939-f001:**
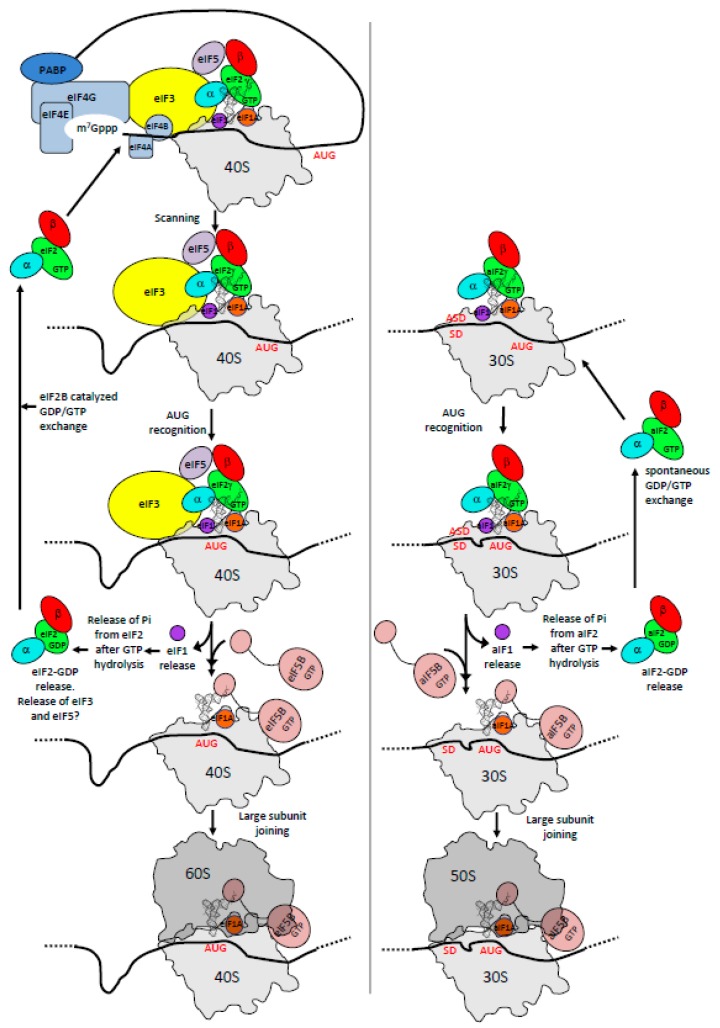
Schematic views of the translation initiation steps in eukaryotes and in archaea. The figure illustrates the main steps in the formation of the pre-initiation complex (PIC) in eukaryotes (left) and in archaea (right). The translation competent IC is formed after the release of e/aIF1A and e/aIF5B. The complex formed by eIF4E + eIF4G + eIF4A is known as eIF4F. Note that the e/aIF2 heterotrimer is represented with a three-color code (α subunit in blue, β subunit in red, γ subunit in green) for consistency with the other figures, highlighting the role of each subunit in the translation initiation. eIF3, composed of 6 (yeast) to 13 (mammals) subunits is represented as a single yellow oval to simplify the figure.

**Figure 2 ijms-20-00939-f002:**
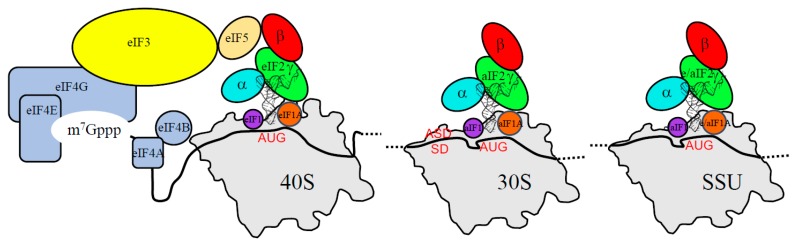
Definition of a common structural core for the start codon selection in eukaryotes and in archaea. The figure highlights the common elements used in eukaryotes and archaea for the key steps of the start codon selection. Left, eukaryotic PIC; middle, archaeal PIC; and right, common core.

**Figure 3 ijms-20-00939-f003:**
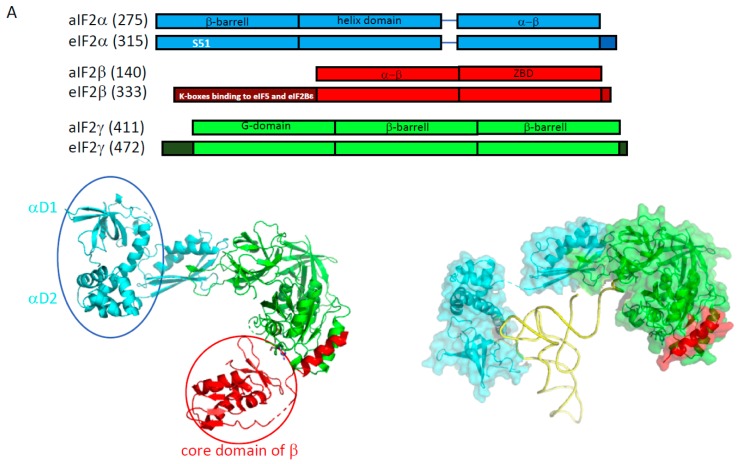
The structure of the initiation factors belonging to the core complex. (**A**) The structural organization of the three subunits of e/aIF2 is shown. Overall, aIF2α, β and γ from *P. abyssi* have about 28%, 40%, and 48% sequence identity with the eukaryotic eIF2α, β and γ subunits, respectively. The eukaryotic specificities of each factor are schematized by dark boxes or indicated within the block with white letters. The eukaryotic eIF2α have an acidic C-terminal extension. The eukaryotic eIF2β have a supplementary N-terminal domain necessary for the binding to eIF5 and eIF2Bε. The eukaryotic eIF2γ have N- and C-terminal extensions compared to the archaeal versions. Note that some eukaryotic and/or archaeal specificities are also found at some positions in the sequences (see text and [66,67,68] for sequence alignments). The structure of the full heterotrimeric aIF2 is shown in a cartoon representation in its unbound form (left, PDB ID 2AHO and 2QMU) and bound to Met-tRNA_i_^Met^ (right, PDB ID 3V11). The α subunit is colored in cyan, the β subunit is colored in red and the γ subunit is colored in green. The two mobile wings of aIF2, the αD12 domains and the core domain of β, are circled. (**B**) The structures of e/aIF1A and of its bacterial homologue IF1. The structure of aIF1A from *P. abyssi* (PDB ID 4MNO) is drawn in orange cartoons. Overall, eIF1A from *P. abyssi* has about 40% identity with the eukaryotic eIF1A. The basic and acidic eukaryotic extensions necessary for long-range scanning are shown with light yellow boxes. The archaeal version of the factor contains a helix domain at the C-terminal extremity but no acidic extension. However, a short N-terminal tail is present in aIF1A. The organization of the bacterial IF1 protein is also shown. As described in the text, because of their structural resemblance, the IF1 and e/aIF1A proteins are considered as universal initiation factors. The secondary structures are from *E. coli* IF1 (PDB code 1AH9) and from *P. abyssi* aIF1A (PDB code 4MNO). (**C**) Structure of e/aIF1. The structure of aIF1 from *Methanocaldococcus jannaschii* (PDB ID 4MO0) is drawn in magenta cartoons, and that of eIF1 from *S. cerevisiae* (PDB ID 2OGH) is in purple. Overall, eIF1 from *P. abyssi* has about 30% sequence identity with the eukaryotic eIF1. The L1 loop, also called a basic loop, is located in front of the anticodon loop of the tRNA in the P_IN_ states of eukaryotes and archaea. The loop is mobile outside of the ribosome (Figure 4). The L2 loop is only present in eukaryotes [72]. It contacts the tRNA D-loop in the P_IN_ state [73]. In all the panels, the numbers of residues indicated in parentheses refer to *P. abyssi* proteins for the archaea, human proteins for eukaryotes, and *E. coli* for IF1.

**Figure 4 ijms-20-00939-f004:**
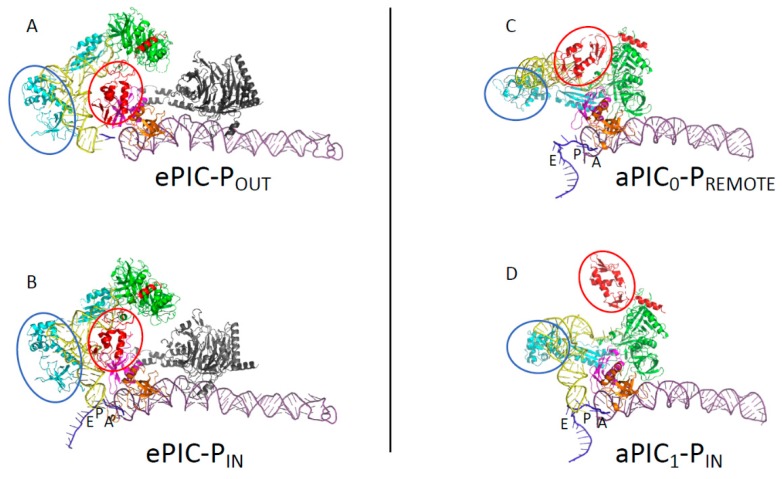
A comparison of archaeal and eukaryotic initiation complexes. The figure shows partial views (P site region) of eukaryotic and archaeal initiation complexes. (**A**) The eukaryotic 48S-open P_OUT_ complex. (**B**) The eukaryotic 48S-closed P_IN_ complex. (**A**) and (**B**) are from [134] (PDB ID 3JAQ, 3JAP). (**C**) The archaeal PIC0-P_REMOTE_ conformation. (**D**) The archaeal PIC1-P_IN_ conformation. The color code for the initiation factors is the same as in Figure 3. The small ribosomal helix h44 is in dark purple and the mRNA is in dark blue. The eIF3 subunits in views (**A**) and (**B**) are in black. The two mobile wings of e/aIF2, as defined in the legend of Figure 3, are encircled. Archaeal aIF1 and aIF1A have positions similar to those of their eukaryotic orthologues eIF1 and eIF1A. In eukaryotes, the initiator tRNA is bound to the γ subunit of eIF2 and to the domain 3 of eIF2α, as observed in the archaeal TC [69]. However, eIF2αD12 has moved and is found in the E site, while the core domain of β is close to the tRNA. No interaction is observed between the γ subunit and h44. In archaea, the structure of the TC bound to the ribosome corresponds to that observed outside the ribosome for the IC0-P_REMOTE_ complex. The γ subunit contacts h44 and aIF1. In the IC1-P_IN_ conformation, the structure of the TC is constrained, but the energetic cost would be compensated by the codon:anticodon base pairing. The movement of the two mobile wings of aIF2 may help the start codon selection.

**Figure 5 ijms-20-00939-f005:**
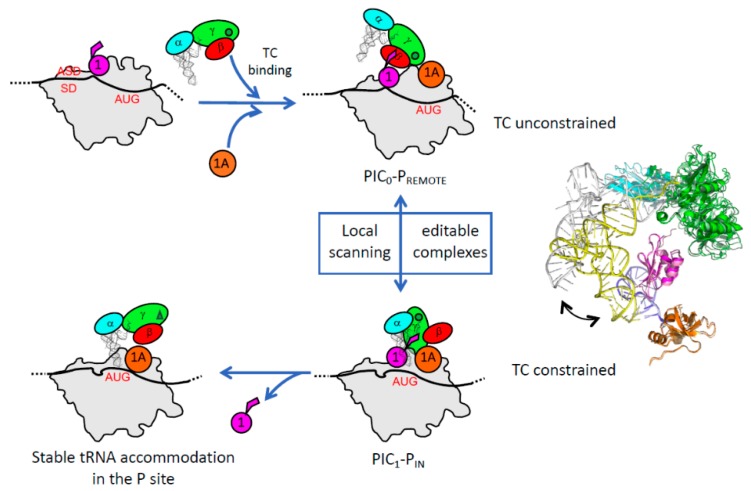
Local scanning for the start codon selection in archaea. The figure schematizes the different steps of the start codon selection for SD containing archaeal mRNAs. Note that aIF1 facilitates the mRNA binding [72,145]. On the right of the figure, the positions of the TC in PIC0–P_REMOTE_ and PIC1–P_IN_ are compared after superimposition of the 30S bodies. The color code is the same as that in Figure 3 and Figure 4, with light colors for PIC0–P_REMOTE_ and dark colors for PIC1–P_IN_. For clarity, the mobile wings of aIF2 are omitted. The view shows that aIF2γ does not significantly move in the two conformations, leading to a structural constraint in PIC1-P_IN_.

**Table 1 ijms-20-00939-t001:** The main features of the translation initiation in the three domains of life. The main mRNA, initiator tRNA features, and correspondence between the initiation factors in the three domains of life are shown. *: IF3 is a two-domain protein. The correspondence between IF3 and e/aIF1 is based on a structural and functional resemblance of the IF3 C-terminal domain with e/aIF1. Despite this resemblance, the topologies of the two α–β folds are different. This suggests a convergent evolution rather than the occurrence of a common ancestor. ^§^: It should be underlined that, because of its homology with the bacterial IF2, aIF5B was misleadingly called aIF2 in some early publications. ^#^: The catalytic γ and ε subunits of eIF2B are missing in archaea. The function of the eIF2B α,β,δ homologues in archaea is not clear [43,44]. ^!^: aIF4A present in some archaea.

Eukaryotes	Archaea	Bacteria
**mRNA features**		
Canonical capped dependent, Kozak motif Non-canonical capped dependent Non-canonical capped independent IRES-mediated Leaderless	SD-dependent Leaderless	SD-dependent Leaderless
**main initiator tRNA features**		
methionine A_1_-U_72_ G_29_-C_41_, G_30_-C_40_, G_31_-C_39_	methionine A_1_-U_72_ G_29_-C_41_, G_30_-C_40_, G_31_-C_39_	formyl-methionine mispaired 1-72 bases G_29_-C_41_, G_30_-C_40_, G_31_-C_39_
**Translation initiation factors**		
eIF2 (α,β,γ)	aIF2 (α,β,γ)	-
eIF1	aIF1	~IF3 *
eIF1A	aIF1A	IF1
eIF5B ^§^	aIF5B ^§^	IF2
eIF5	-	-
eIF2B (α,β,γ,δ,ε)	aIF2B (α,β,δ) ^#^	-
eIF3 (6 to 13 subunits)	-	-
eIF4F (4A, 4G, 4E)	aIF4A ^!^	-
eIF4B	-	-

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
