# Peer review of "Start Codon Recognition in Eukaryotic and Archaeal Translation Initiation: A Common Structural Core"

_ijms, 2019, doi:10.3390/ijms20040939_

Round 1
Reviewer 1 Report
Present review is dedicated to comparison of the initiation step of translation in eukaryotes and archea. The authors meticulously describe the well-known and new facts about pre-initiation complex formation, functions of some initiation factors, and correct methionine initiator tRNA positioning. Moreover, these processes are considered in an evolutionary context, that gives a special interest to the article. Nevertheless, some structural data are described somewhat superficially, and are a bit difficult to understand by a non-specialist.
This impressive review covers a wealth of data available to date, is written in a good scientific language, and carefully illustrated. In my opinion, this work can be published in the journal after minor revision - spell checking and some language-style polishing.
Author Response
We thank the reviewers for meticulous assessment of our work. Their detailed suggestions were very helpful to improve our manuscript and the revised version carefully takes into account all referees’ comments.
We provide below our point-by-point responses to the referees’ comments, which detail our changes in the manuscript. To provide a quick overview of our revisions, we have highlighted in red all new or altered portions of text in the revised manuscript.
Present review is dedicated to comparison of the initiation step of translation in eukaryotes and archea. The authors meticulously describe the well-known and new facts about preinitiation complex formation, functions of some initiation factors, and correct methionine initiator tRNA positioning. Moreover, these processes are considered in an evolutionary context, that gives a special interest to the article. Nevertheless, some structural data are described somewhat superficially, and are a bit difficult to understand by a non-specialist. This impressive review covers a wealth of data available to date, is written in a good scientific language, and carefully illustrated. In my opinion, this work can be published in the journal after minor revision - spell checking and some language-style polishing.
We thank the reviewer for these comments. The text was read by a native English speaker and
corrected accordingly.
Reviewer 2 Report
Comments for authors:
The authors discuss the latest developments in the understanding of translation initiation in Archaea, and contrast it with the processes in Eukarya (and Bacteria). In addition to reviewing the literature on the biochemical and structural studies of translation initiation, they also place the mechanistic understanding in an evolutionary context that is motivated by the latest developments in inferring the evolutionary relationships of the major branches of the tree of life. In doing so they raise an interesting question about leaderless mRNA translation as to whether such processes of 'direct' recognition of start codon is an ancestral relic or a recently evolved mechanism. The English is good overall, although I noticed some sentences that sound odd grammatically, probably because of a somewhat uncommon usage of some adjectives. In general, this is a useful review that summarizes the current knowledge of archaeal translation initiation as well as the evolutionary implications of the new understanding. However, the literature review is neither thorough nor up-to-date as far as the evolutionary aspects relevant to the discussion are concerned. Suggestions to improve the manuscript with regards to what I see as deficiencies in the completeness of literature review (major) as well as grammatical issues (minor) are described below.
Suggestions for improving the review:
Major issues:
(1) I think that it will be useful for a broad range of readers if a more general description (and comparison) of the diversity of mechanisms of initiation and start codon recognition is provided in the introduction. For example, with a focus on features of the mRNA in Archaea, Bacteria and Eukarya: leaderless and capped mRNAs, IRES motifs as well as Kozak motifs in addition to Shine Dalgarno motifs. This would be useful to motivate the discussion both in mechanistic and evolutionary contexts. Perhaps, Table 1 can be enhanced with a comparison of mRNA features to show the correspondence with the IFs in Archaea, Bacteria and Eukarya. Such an enhanced Table 1 will not only better complement Fig. 2 in my opinion, but it would also better relate to Fig. 2 to reflect the repertoires of IFs in different organisms.
(2) The three-domains of life model has been challenged by two alternative two-domains models: (1) Eocyte model in which archaea are paraphyletic (split into more than one group) and (2) Akaryote model, in which the Archaea are monophyletic and siter to Bacteria. Both the two-domains models have received support from improved phylogenomic analyses recently. Note that the Akaryote model is the much better supported than the Eocyte model, both statistically as well as from a structural biology perspective (see Refs 4-6 below). However, the authors have ignored the second model completely. This is probably due to unfamiliarity rather than a preference/prejudice for specific evolutionary models.
The authors should fill the deficiency and revise the manuscript to reflect the better supported two-domains model and the implications for the evolution of translation. Some relevant literature is listed below, but more relevant articles can also be found in the references therein. Further, the references suggested below discuss a more general, and important process of evolution -- Reductive evolution. These references are particularly relevant to the discussion in section 1 (introduction/motivation) and section 4 (nature of the translation mechanisms in LUCA) of the manuscript. With regards to section 4, see also minor comments.
1. Brinkmann, H. and H. Philippe (1999). Archaea sister group of Bacteria? Indications from tree reconstruction artifacts in ancient phylogenies. Molecular biology and evolution 16(6): 817-825.
2. Forterre, P. and H. Philippe (1999). Where is the root of the universal tree of life? BioEssays 21(10): 871-879.
3. Lecompte, O., R. Ripp, J. C. Thierry, D. Moras and O. Poch (2002). Comparative analysis of ribosomal proteins in complete genomes: An example of reductive evolution at the domain scale. Nucleic Acids Research 30(24): 5382-5390.
4. Kurland, C. G. and A. Harish (2015). Structural biology and genome evolution: An introduction. Biochimie 119: 205-208.
5. Harish, A. and C. G. Kurland (2017). Akaryotes and Eukaryotes are independent descendants of a universal common ancestor. Biochimie 138: 168-183.
6. Harish, A. (2018). What is an archaeon and are the Archaea really unique? PeerJ 6: e5770.
Lines 30-44: It is common knowledge that tertiary structure of macromolecules is much more conserved than primary structure (i.e., sequence), and thus tertiary structure evolves slowly compared to primary structure. Accordingly, it is becoming increasingly clear that structure-based approaches of deep phylogenetic studies are more robust than traditional sequence-based approaches (refs 4-6). Therefore structure-based approaches to phylogenetic studies are more reliable.
Minor issues:
Lines 50-51: In its current form, the statement sounds like a vague, unsubstantiated opinion. Explaining how studies in archaea will inform eukaryotic models with an example will be useful, rather than simply citing an article. In any case, Ref 21 is a bit too much of a stretch to claim archael models inform models for human disease.
Line 98: A sentence or two in 2.1, to motivate what follows in 2.1.1 - 2.1.3 will be better than abruptly starting with 2.1.1.
Line 158: Homology is an inference based on extensive similarity. Therefore the usage 'Structural homology' is inappropriate and cryptic. Even if it is the common usage, it is misleading and should be revised. This should be better described here, and also at other instances, if any.
Lines 403-412: The latter half of the final paragraph starting with "Finally, diveristy in possible ..." seems somewhat cryptic. Do the authors mean to say that studies of molecular mechanisms can complement phylogenetic studies or that insights from molecular mechanisms could go beyond phylogenetic studies and provide meta-information? This issue should be clarified and elaborated. Accordingly, this can be a separate (and concluding) paragraph.
Author Response
We thank the reviewers for meticulous assessment of our work. Their detailed suggestions were very helpful to improve our manuscript and the revised version carefully takes into account all referees’ comments.
We provide below our point-by-point responses to the referees’ comments, which detail our changes in the manuscript. To provide a quick overview of our revisions, we have highlighted in red all new or altered portions of text in the revised manuscript.
Comments for authors:
The authors discuss the latest developments in the understanding of translation initiation in Archaea, and contrast it with the processes in Eukarya (and Bacteria). In addition to reviewing the literature on the biochemical and structural studies of translation initiation, they also place the mechanistic understanding in an evolutionary context that is motivated by the latest developments in inferring the evolutionary relationships of the major branches of the tree of life. In doing so they raise an interesting question about leaderless mRNA translation as to whether such processes of 'direct' recognition of start codon is an ancestral relic or a recently evolved mechanism. The English is good overall, although I noticed some sentences that sound odd grammatically, probably because of a somewhat uncommon usage of some adjectives. In general, this is a useful review that summarizes the current knowledge of archaeal translation initiation as well as the evolutionary implications of the new understanding. However, the literature review is neither thorough nor up-to-date as far as the evolutionary aspects relevant to the discussion are concerned. Suggestions to improve the manuscript with regards to what I see as deficiencies in the completeness of literature review (major) as well as grammatical issues (minor) are described below.
We thank the reviewer for these comments. The text was read by a native English speaker and corrected accordingly. The reviewer's suggestions have been carefully taken into account and the corrections detailed below have been made.
Suggestions for improving the review:
Major issues:
(1) I think that it will be useful for a broad range of readers if a more general description (and comparison) of the diversity of mechanisms of initiation and start codon recognition is provided in the introduction. For example, with a focus on features of the mRNA in Archaea, Bacteria and Eukarya: leaderless and capped mRNAs, IRES motifs as well as Kozak motifs in addition to Shine Dalgarno motifs. This would be useful to motivate the discussion both in mechanistic and evolutionary contexts. Perhaps, Table 1 can be enhanced with a comparison of mRNA features to show the correspondence with the IFs in Archaea, Bacteria and Eukarya. Such an enhanced Table 1 will not only better complement Fig. 2 in my opinion, but it would also better relate to Fig. 2 to reflect the repertoires of IFs in different organisms.
According to the reviewer's suggestions, we enhanced Table 1 and the description of eukaryotic translation initiation mechanisms in the Introduction section (end of the second paragraph of the Introduction).
(2) The three-domains of life model has been challenged by two alternative two-domains models: (1) Eocyte model in which archaea are paraphyletic (split into more than one group) and (2) Akaryote model, in which the Archaea are monophyletic and siter to Bacteria. Both the two-domains models have received support from improved phylogenomic analyses recently. Note that the Akaryote model is the much better supported than the Eocyte model, both statistically as well as from a structural biology perspective (see Refs 4-6 below). However, the authors have ignored the second model completely. This is probably due to unfamiliarity rather than a preference/prejudice for specific evolutionary models.
The authors should fill the deficiency and revise the manuscript to reflect the better supported twodomains model and the implications for the evolution of translation. Some relevant literature is listed below, but more relevant articles can also be found in the references therein. Further, the references suggested below discuss a more general, and important process of evolution -- Reductive evolution. These references are particularly relevant to the discussion in section 1 (introduction/motivation) and section 4 (nature of the translation mechanisms in LUCA) of the manuscript. With regards to section 4, see also minor comments.
1. Brinkmann, H. and H. Philippe (1999). Archaea sister group of Bacteria? Indications from tree reconstruction artifacts in ancient phylogenies. Molecular biology and evolution 16(6): 817-825.
2. Forterre, P. and H. Philippe (1999). Where is the root of the universal tree of life? BioEssays 21(10): 871-879.
3. Lecompte, O., R. Ripp, J. C. Thierry, D. Moras and O. Poch (2002). Comparative analysis of ribosomal proteins in complete genomes: An example of reductive evolution at the domain scale. Nucleic Acids Research 30(24): 5382-5390.
4. Kurland, C. G. and A. Harish (2015). Structural biology and genome evolution: An introduction. Biochimie 119: 205-208.
5. Harish, A. and C. G. Kurland (2017). Akaryotes and Eukaryotes are independent descendants of a universal common ancestor. Biochimie 138: 168-183.
6. Harish, A. (2018). What is an archaeon and are the Archaea really unique? PeerJ 6: e5770.
Lines 30-44: It is common knowledge that tertiary structure of macromolecules is much more conserved than primary structure (i.e., sequence), and thus tertiary structure evolves slowly compared to primary structure. Accordingly, it is becoming increasingly clear that structure based approaches of deep phylogenetic studies are more robust than traditional sequence based approaches (refs 4-6). Therefore structure-based approaches to phylogenetic studies are more reliable.
We added large portions of text in the Introduction in order to better describe the other models suggested by the reviewer. The references have also been updated, as requested (see Introduction, first paragraph).
Minor issues:
Lines 50-51: In its current form, the statement sounds like a vague, unsubstantiated opinion. Explaining how studies in archaea will inform eukaryotic models with an example will be useful, rather than simply citing an article. In any case, Ref 21 is a bit too much of a stretch to claim archael models inform models for human disease.
Several references reporting studies showing that archaeal proteins give valuable information on eukaryotic processes are now cited (References 29 to 34 of the revised versions).
Line 98: A sentence or two in 2.1, to motivate what follows in 2.1.1 - 2.1.3 will be better than abruptly starting with 2.1.1.
A paragraph now introduces section 2.1
Line 158: Homology is an inference based on extensive similarity. Therefore the usage 'Structural homology' is inappropriate and cryptic. Even if it is the common usage, it is misleading and should be revised. This should be better described here, and also at other instances, if any.
The phrase 'Structural homology' has been replaced by "Structural resemblance".
Lines 403-412: The latter half of the final paragraph starting with "Finally, diveristy in possible ..." seems somewhat cryptic. Do the authors mean to say that studies of molecular mechanisms can complement phylogenetic studies or that insights from molecular mechanisms could go beyond phylogenetic studies and provide meta-information? This issue should be clarified and elaborated. Accordingly, this can be a separate (and concluding) paragraph.
A separate concluding paragraph is present in the revised version. It is now more clearly explained that studies of molecular mechanisms can complement phylogenetic studies. New examples are also given.

Reviewer 3 Report
Generally, I think this is a sound review that makes an interesting and relevant comparison of archaeal and eukaryotic translation initiation.
My biggest concerns are:
1 - They need to polish the writing. Poor grammar or style interrupted my concentration.
2 - They understandably want to focus on the "core" factors that are common between eukaryotes and archaea, but this sometimes causes them to gloss over (or outright ignore) other important factors and their functions.
Specific comments:
Why do authors use “e.g.” just before references? I don’t think that it is a right use of “e.g.”.
Line # 30: Don't they mean 18S for eukaryotes? I understand their point about slow evolution, but that doesn't mean they haven't changed at all.
Figure 3 (legend): From which organism? Yeast? Humans? It isn't clear.
Line # 203: It would be nice if they mention Stephen Lee's Cell Reports paper somewhere since his work argues that, at least under hypoxia, a/eIF5B carries out the tRNA-delivery role. I wonder if this occurs in archaea even in the absence of eIF2-phosphorylation.
Line # 245: either say "TC" or "ternary complex".
Line # 306: I think Lin et al (2018) would argue that the dynamic exchange of eIF5:eIF5B for eIF1A:eIF5B upon eIF2-GTP hydrolysis, and the subsequent (or possibly coupled) release of eIF5:eIF2-GDP, is important here.
Line # 370: Again, authors are ignoring the potential importance of aIF5B, at least for the translation of hypoxia-response elements.
uORF-mediated translation, which operates under stress conditions, has been referenced in the manuscript. However, there is hardly any mention about cellular IRES-mediated translation that operates under eIF2alpha phosphorylation conditions.
Author Response
We thank the reviewers for meticulous assessment of our work. Their detailed suggestions were very helpful to improve our manuscript and the revised version carefully takes into account all referees’ comments.
We provide below our point-by-point responses to the referees’ comments, which detail our changes in the manuscript. To provide a quick overview of our revisions, we have highlighted in red all new or altered portions of text in the revised manuscript.
Generally, I think this is a sound review that makes an interesting and relevant comparison of archaeal and eukaryotic translation initiation.
We thank the reviewer for these comments. The text was read by a native English speaker and corrected accordingly. The reviewer's suggestions have been carefully taken into account and the corrections detailed below have been made.
My biggest concerns are:
1 - They need to polish the writing. Poor grammar or style interrupted my concentration.
The text was read by a native English speaker and corrected accordingly.
2 - They understandably want to focus on the "core" factors that are common between eukaryotes and archaea, but this sometimes causes them to gloss over (or outright ignore) other important factors and their functions.
Alternative routes for translation initiation in eukaryotes and the associated factors are now cited in the Introduction and in section 2.1.2 (end of the first paragraph). In addition, the role of e/aIF5B is now better documented (see below).
Specific comments:
Why do authors use “e.g.” just before references? I don’t think that it is a right use of “e.g.”.
e.g has been removed throughout the text.
Line # 30: Don't they mean 18S for eukaryotes? I understand their point about slow evolution, but that doesn't mean they haven't changed at all.
We have now stated 16S/18S to take into account eukaryotes.
Figure 3 (legend): From which organism? Yeast? Humans? It isn't clear.
The organisms were indicated at the end of the legend, and therefore it was not clear that this indication was valid for all panels. This is now clearly stated.
Line # 203: It would be nice if they mention Stephen Lee's Cell Reports paper somewhere since his
work argues that, at least under hypoxia, a/eIF5B carries out the tRNA-delivery role. I wonder if this occurs in archaea even in the absence of eIF2-phosphorylation.
Section 2.1.2 has been updated to mention alternative routes of translation initiation, in particular those where eIF5B is used as a tRNA carrier. Moreover, a sentence has been added at the end of the section to leave open the possible existence of such a role in archaea.
Line # 245: either say "TC" or "ternary complex".
This has been corrected.
Line # 306: I think Lin et al (2018) would argue that the dynamic exchange of eIF5:eIF5B for
eIF1A:eIF5B upon eIF2-GTP hydrolysis, and the subsequent (or possibly coupled) release of
eIF5:eIF2-GDP, is important here.
A sentence citing this article has been added at the end of section 3.1.
Line # 370: Again, authors are ignoring the potential importance of aIF5B, at least for the translation of hypoxia-response elements.
The sentence only concerns the cases where the heterotrimeric e/aIF2 is used. To clarify, we replaced "involving" by "that involves".
uORF-mediated translation, which operates under stress conditions, has been referenced in the
manuscript. However, there is hardly any mention about cellular IRES-mediated translation that
operates under eIF2alpha phosphorylation conditions.
These points are now mentioned in the Introduction section (end of second paragraph) and in Table 1.
